# Prevalence of EBV, HHV6, HCMV, HAdV, SARS-CoV-2, and Autoantibodies to Type I Interferon in Sputum from Myalgic Encephalomyelitis/Chronic Fatigue Syndrome Patients

**DOI:** 10.3390/v17030422

**Published:** 2025-03-14

**Authors:** Ulf Hannestad, Annika Allard, Kent Nilsson, Anders Rosén

**Affiliations:** 1Department of Biomedical & Clinical Sciences, Division of Cell & Neurobiology, Linköping University, SE-58185 Linköping, Sweden; ulf.hannestad@gmail.com; 2Department of Clinical Microbiology, Clinical Virology, Umeå University, SE-90185 Umeå, Sweden; annika.allard@regionvasterbotten.se; 3Department of Pain and Rehabilitation, Linköping University Hospital, SE-58758 Linköping, Sweden; kent.nilsson@regionostergotland.se

**Keywords:** Epstein–Barr virus, human adenovirus, human herpesvirus 6, human cytomegalovirus, autoAbs to interferon type I, myalgic encephalomyelitis/chronic fatigue syndrome

## Abstract

An exhausted antiviral immune response is observed in myalgic encephalomyelitis/chronic fatigue syndrome (ME/CFS) and post-SARS-CoV-2 syndrome, also termed long COVID. In this study, potential mechanisms behind this exhaustion were investigated. First, the viral load of Epstein–Barr virus (EBV), human adenovirus (HAdV), human cytomegalovirus (HCMV), human herpesvirus 6 (HHV6), and severe acute respiratory syndrome coronavirus 2 (SARS-CoV-2) was determined in sputum samples (n = 29) derived from ME/CFS patients (n = 13), healthy controls (n = 10), elderly healthy controls (n = 4), and immunosuppressed controls (n = 2). Secondly, autoantibodies (autoAbs) to type I interferon (IFN-I) in sputum were analyzed to possibly explain impaired viral immunity. We found that ME/CFS patients released EBV at a significantly higher level compared to controls (*p* = 0.0256). HHV6 was present in ~50% of all participants at the same level. HAdV was detected in two cases with immunosuppression and severe ME/CFS, respectively. HCMV and SARS-CoV-2 were found only in immunosuppressed controls. Notably, anti-IFN-I autoAbs in ME/CFS and controls did not differ, except in a severe ME/CFS case showing an increased level. We conclude that ME/CFS patients, compared to controls, have a significantly higher load of EBV. IFN-I autoAbs cannot explain IFN-I dysfunction, with the possible exception of severe cases, also reported in severe SARS-CoV-2. We forward that additional mechanisms, such as the viral evasion of IFN-I effect via the degradation of IFN-receptors, may be present in ME/CFS, which demands further studies.

## 1. Introduction

Myalgic encephalomyelitis/chronic fatigue syndrome (ME/CFS) is a complex, often debilitating, chronic disease that affects around 3.3 million people in the United States as estimated by the Center for Disease Control and Prevention, USA, in 2021–2022 [1]. The number of people affected by ME/CFS worldwide was estimated to be about 65 million before the COVID-19 pandemic [2]. Patients suffering from ME/CFS experience post-exertional malaise, severe fatigue, cognitive disturbances/brain fog, sleep and immunological dysfunctions, with a pronounced impact on daily quality of life [3]. The biomolecular mechanisms behind ME/CFS are still unknown. However, the onset of ME/CFS in most cases occurs after an episode with flu-like symptoms, indicating that infectious agents can play a role in the continuing symptoms of the disease [4]. EBV-induced infectious mononucleosis is often reported at the onset of ME/CFS, as well as other infectious agents, i.e., Ross River virus, *Coxiella burnetii*, West Nile virus, and SARS-CoV-2 [3,5,6]. Non-infectious events, such as physical or mental stress, toxins, or trauma, are known triggers of ME/CFS [7].

Following the occurrence of the SARS-CoV-2 pandemic, the prevalence of patients suffering from prolonged symptoms lasting for more than 6 months, e.g., long COVID, is estimated to affect between 5% and 43% of infected patients. Most long COVID symptoms are also seen in ME/CFS, and it is difficult to distinguish between these two conditions [8].

We previously showed that, in SARS-CoV-2 infected persons, the antibody titers in saliva against EBV and HHV6 were significantly increased, indicating a reactivation of these viruses after an infection trauma such as COVID-19 [9]. The anti-EBV mucosal Abs were more enhanced in ME/CFS patients compared to healthy donors. In a recent study, we also showed that the concentration of anti-HAdV antibodies in saliva was increased after infection with SARS-CoV-2 [10]. These findings raised the question of which role HAdV and/or herpesviruses play in the pathogenesis of ME/CFS.

EBV, HCMV, HHV6, SARS-CoV-2, and HAdV all replicate in the airway epithelium [11,12,13]. These viruses are transmitted via saliva, sputum, or via the inhalation of droplets from infected individuals, and can establish persistent infections in their host, in part through evading host immune surveillance [14]. The efficacy of innate immunity, particularly the interferon-mediated response, is critical to control viral infections at an early stage and to trigger at a later stage a broad spectrum of specific adaptive immune responses against i.e. EBV [14]. The virus governs innate immune responses of its host in various ways, including the degradation of IFN-α receptors IFNAR1 and IFNR2, and exploits them to their advantage [14]. It is increasingly realized that individuals with chronic autoimmune diseases develop neutralizing autoAbs against IFN-I, thereby compromising their own innate antiviral defenses [15]. Patients with ME/CFS have enhanced susceptibility for viral diseases, particularly EBV [7], but the underlying mechanisms, such as dysfunctional IFN-I responses, are not understood.

In this study, we investigate the presence and viral load of EBV, HCMV, HHV6, HAdV, and SARS-CoV-2 in airway mucosa by analyzing sputum samples derived from ME/CFS patients and controls. We hypothesize that dysfunctional IFN signaling underlies the aberrant control of viral infection by analyzing autoAbs to IFN-I.

## 2. Materials and Methods

### 2.1. Participants

Sputum samples were taken from patients (n = 13, median age 52.5 yrs, range of 22–61 yrs), who were recruited from the regional (Östergötland) Association for ME patients in Sweden. They fulfilled the ME/CFS diagnosis according to the 2003 Canadian Consensus Criteria [16,17], including fatigue with post-exertional malaise (criteria 1), sleep disturbances (criteria 2), pain (criteria 3), neurological /cognitive manifestations (criteria 4), autonomous /neuroendocrine /immune manifestations (criteria 5), and a duration of illness for at least 6 months (criteria 6). Information related to ME/CFS trigger events (infection, trauma, stress, vaccination, or other), disease duration, and past infections, were retrieved via self-reported questionnaire. The inclusion criteria for the healthy control group were active working donors, matching age of patients, and devoid of ME/CFS diagnosis. The exclusion criteria for participant enrollment were existence of current active infection and/or infectious disease symptoms and age below 18 years. Thus, participants had no evidence of active SARS-CoV-2 or other infection at the time of sampling. Disease severity in patients with ME/CFS was assessed by a physician on a 1 (mild) to 4 (very severe) scale: 1 (mild): approximately 50% reduction in activity; 2 (moderate): mostly housebound; 3 (severe) mostly bedbound; and 4 (very severe): bedbound and dependent on help for physical functions. The median duration of ME/CFS disease in this group was 12 years, with a range of 8–28 years. Sputum samples were also collected from healthy age-matched donors (HD) (n = 10, median age 52.5 yrs, range 33–66 yrs). The age of ME/CFS vs. HD did not differ (*p* = 0.7089) (Table 1). Four elderly healthy donors (Senior control donors, SENIORS) were also recruited; median age 73.5 yrs, range 65–77 yrs, as well as two immunosuppressed participants: one B-cell-depleted donor (Rituximab anti-CD19 mAb treated, named NEG CTR, based on the low count of Ab-releasing B-cells and EBV), and one glucocorticoid immunosuppressed donor (treatment of asthma, named POS CTR, based on the presence of all analyzed viruses except SARS-CoV-2). The POS and NEG CTR donors were both active and working at 100% capacity. Demographic data, including sex, age, and duration of ME/CFS disease, of the study participants are presented in Table 1.

### 2.2. Ethical Permit

This study was reviewed and approved by the Swedish Ethical Review Authority, Regional Ethics Committee (D.nr. 2019-0618 and 2024-00365-02). Health declarations, medical data, and sputum samples were collected after written informed consent.

### 2.3. Sputum Collection

Sputum samples were collected in the morning according to detailed instructions, including twice rinsing the mouth with water and deep breathing. The mucus sputum sample was then collected by strong coughing of sputum into a sterile 50 mL plastic tube, trying to avoid saliva contamination. Sputum samples were diluted with an equal volume 0.9% NaCl solution, mixed vigorously for 2 min to reduce viscosity and facilitate handling, and then frozen at −80 °C.

### 2.4. Analysis of EBV, HCMV, HHV6, HAdV, and SARS-CoV-2 in Sputum

All samples were analyzed for HAdV [18], EBV, HCMV [19], HHV6 (forward primer 5′-GCG TTT TCA GTG TGT AGT TCG G-3′, reverse primer 5′-TTC TGT GTA GGC GTT TCG ATC A-3′, probe 5′-FAM--CCT CAA CCT AGC GCT CGG GGC T-TAMRA-3′; unpublished), and SARS-CoV-2 (modified from Corman et al. [20]) using a real-time PCR at the Department of Clinical Microbiology, Umeå University. Nucleic acid (DNA/RNA) was prepared from sputum diluted with equal volume of isotonic NaCl, followed by routine extraction using the QIASymphony^®^ DSP Virus/Pathogen Midi Kit (QIAgen, Hilden, Germany). Detection and quantification were performed on a real-time PCR instrument QuantStudio5 (AppliedBiosystems by Thermofisher Scientific Inc., Waltham, MA, USA). For construction of an in-house standard to be used for quantification, plaque formation studies of HSV1, HSV2, and adenovirus were performed together with TCID_50_ (50% tissue culture infectious dose) dilutions of infected cells. In addition, these results were combined with data of spectrophotometrically measured pure HSV1- and adenovirus DNA. Taken together, these results made it possible to construct a standard curve with HSV2 as a template with a linear range of 5–500,000 copies adapted to double-stranded DNA viruses. The credibility of the quantitative DNA standard to be used for all DNA viruses included in this study was confirmed by external panels from QCMD (Quality Controls for Molecular Diagnostics, Glasgow, UK). Amplirun (Vircell molecular, Granada, Spain) was used as an external control for SARS-CoV-2.

### 2.5. ELISA Analysis of IgG autoAbs to Type-I IFN

The assay was performed as previously described [21]. In brief, ELISA was performed in 96-well ELISA plates (Maxisorp; Thermo Fisher Scientific, Inc., Waltham, MA, USA) coated overnight in 4 °C with 1 µg/mL recombinant human IFN-α2 (ref. number 130-093-873; Miltenyi Biotec, Tokyo, Japan). The plates were washed in PBS with 0.05% Tween 20, blocked by assay buffer (PBS, 0.05% Tween 20, 0.5% BSA), and washed and incubated with a 1:16 dilution of sputum samples from ME/CFS and control groups for 2 h at room temperature. Each sample was tested in duplicate. Plates were washed with PBS, 0.05% Tween 20, and incubated with a goat anti-human IgG (Fc)-HRP conjugate (Bio-Rad, Tokyo, Japan) at a 1:5000 dilution for 1 h at room temperature. After a final wash in PBS, 0.05% Tween 20, the TMB substrate was added, and the optical density was measured at 450 nm.

### 2.6. Statistical Analyses

For the statistical analyses of *p*-levels and normality checks, we used JMP version 13.2.1 software (JMP Statistical Discovery Inc., Cary, NC, USA). Statistically significant differences and *p*-levels were calculated according to the non-parametric Wilcoxon rank procedure.

## 3. Results

### 3.1. Viral Load of EBV, HCMV, HHV6, HAdV, and SARS-CoV-2 in Sputum

We found that ME/CFS patients more frequently (85%, 11/13) released EBV compared to HD (50%, 5/10) (Figure 1) and that the viral load, measured as the number of EBV copies/mL, was significantly elevated compared to age-matched controls (*p* = 0.0256) (Figure 2A). EBV copies/mL in the SENIOR vs. HD group showed an elevated trend *p* = 0.0706. The highest EBV load in sputum (526 million copies/mL) was found in the immunosuppressed/glucocorticoid-treated asthmatic donor who inhaled glucocorticoids twice a day. This participant released into sputum all viruses tested, except SARS-CoV-2. In the SENIOR group, all four participants were positive for EBV. The negative control donor (B-cell depleted) was devoid of HAdV, EBV, HHV6, and HCMV (Figure 1). EBV, in its latency state, is harbored in small resting B-cells. This donor was the only one positive for SARS-CoV-2. Notably, both positive and negative control donors were without disease symptoms and were active and working at 100% capacity.

The presence of HHV6 was demonstrated in six out of thirteen patients in the ME/CFS group (Figure 1). In the HD control group, five out of ten were PCR positive for HHV6. In the SENIOR group, two out of four participants were HHV6 positive (Figure 1). There was no statistical difference in viral load of HHV6 in ME/CFS vs. HD. HCMV was not detected in the sputum of any of the 29 participants in this study, except in a glucocorticoid immunosuppressed control donor (Figure 1).

HAdV was detected in a ME/CFS patient (Id12) with severe symptoms, but in none of the HD and SENIOR controls. HAdV was detected, however, in the immunosuppressed/glucocorticoid-treated donor, who was positive for all viruses tested except SARS-CoV-2. Id12 is a patient (age 22 yrs) with severe and long-lasting (12 yrs) ME/CFS. The patient was wheelchair-bound due to severe symptoms. The other ME/CFS participants (12/13) had mild (n = 3), moderate (n = 6), or severe (n = 3) symptoms (Table 1), and none of these were bedridden or wheelchair-bound at the time of sputum collection.

### 3.2. IgG autoAbs Against IFN-I in ME/CFS Patients

On a group level, the autoAbs to IFN-I were at a low basic level, only slightly elevated in ME/CFS patients, albeit not significantly compared to HD controls. Notably, patient Id12 showed somewhat elevated autoAbs to IFN-I compared to the other ME/CFS and HD participants (Figure 3). Additionally, among the SENIORS, the eldest participant (age 77 yrs) showed elevated anti-IFN-I autoAbs. All participants were positive at a low level for anti-type-I IFN IgG autoAbs, except for the negative control (B-cell depleted).

## 4. Discussion

### 4.1. ME/CFS Immune and Antiviral Dysregulation-Unknown Mechanisms

ME/CFS and long COVID are debilitating multisystemic conditions sharing similarities in immune dysregulation and cellular signaling pathways, contributing to a state of immune exhaustion profile in the pathophysiology [22]. These post-acute infection syndromes (PAIS) are subjects of intense research due to the lack of understanding of the underlying mechanisms, representing a significant blind spot in the field of medicine [7,9,23,24,25]. Among the identified risk factors are reactivated latent viruses, including EBV and specific autoAbs [26]. AutoAbs to IFN-I have recently been reported in severe infection: COVID-19 [27,28,29], Ross River virus [30], West Nile virus [21], and in non-SARS-CoV-2 respiratory infections [31]. Notably, these infections are all reported to develop, at a certain frequency, into post-infectious fatigue syndromes sharing most symptoms with ME/CFS [7]. Furthermore, autoAbs to IFN-I are present in higher concentration in approximately 4% of uninfected individuals over 70 years old [28], in autoimmune diseases such as systemic lupus erythematosis (SLE) and Sjögren’s syndrome [15], and in severe of disseminated viral infections caused by Varicella-zoster (VZV), HCMV, or HAdV [32,33]. It is unknown whether autoAbs to IFN-I are present in conditions of lytic viral infections of EBV. Therefore, we hypothesized, in this study, that autoAbs to IFN-I are present in ME/CFS patients—a condition in which oral mucosal Abs against EBV are overexpressed compared to healthy controls [9]. This indicates a higher level/more frequent release of the virus in the mucosa resulting in increased anti-EBV production. However, in contrast to our hypothesis, we found, in the present study, that the majority of the patients and controls expressed low levels of IFN-I Abs, with two notable exceptions: one severely affected ME/CFS patient expressing HAdV and EBV, and one control participant, treated with airway glucocorticoids, expressing HAdV, EBV, HHV6, and HCMV. These findings indicate that, in mild/moderate ME/CFS conditions, the release of EBV and HHV6 is not associated with raised levels of autoAbs to IFN-I. The finding that autoAbs to IFN-I was elevated in a HAdV-positive patient demands follow-up studies in cohorts including patients with severe and/or very severe symptoms. These notions are substantiated by others showing that HAdV and HCMV infections in immunocompromised donors have elevated anti-IFN-I autoAbs that may play a crucial part in down-regulating the antiviral immune defense [15].

### 4.2. Overload of Epstein–Barr Virus in ME/CFS

Several studies have investigated the relationship between EBV and ME/CFS (reviewed in [7,13,33]). For example, Shikova et al. [34] analyzed, in plasma samples, EBV, HHV6, and HCMV among 58 ME/CFS patients, compared to 50 healthy controls, using PCR analysis. They found no significant difference between the two groups regarding the presence of HHV6 or HCMV, whereas a significant increase in lytic/active EBV was detected (*p* = 0.0027). These findings corroborate and are in line with the present results in sputum, where we find a significantly higher number of EBV viral copies (*p* = 0.0256) in ME/CFS patients.

### 4.3. Human Adenovirus and Herpesvirus Reactivation in Airways

Our present study is limited and explorative in nature. The results show that HAdV activation is infrequent and most likely associated with immunosuppression. The POS CTR donor was immunocompromised by glycocorticoids and HAdV-positive. The Id12 patient with severe ME/CFS had active HAdV, possibly due to immunosuppression/exhaustion, resulting in a dysfunctional antiviral surveillance. Therefore, it is of interest to further investigate whether HAdV can have a role in the pathophysiology of ME/CFS, particularly in severe patients. Active HAdV infection is not frequent in the Swedish population (of 32,245 airway infection cases, 2.2% were positive; www.folkhalsomyndigheten.se). Sputum is a superior source for detection of HAdV compared to nasopharyngeal swabs, as reported by Jeong et al. [35]. They found that among 134 patients with respiratory infection, 11 patients in total were positive for HAdV in nasopharyngeal and/or sputum samples. In sputum, the rate of detection was 91%, and in nasopharyngeal swabs, 46% was detected [35].

Both HAdV and herpesviruses establish persistent infections in humans [36]. HAdV persistence can occur in different sites of the body, for example, in T-lymphocytes from tonsils, adenoids, intestine, brain tissue, and in airway epithelial cells [36,37,38]. HAdV was detected in 13 out of 25 brain tissue samples [39]. The authors conclude that the central nervous system can be infected, but it is an overlooked site for HAdV persistent infection. It would be of great importance to analyze whether HAdV infections in the central nervous system could explain the neurological symptoms often seen in ME/CFS. Multiple studies indicate that ME/CFS is initiated and perpetuated by viruses, e.g., herpesviruses and/or possibly HAdV in severe cases, justifying the onset of well-controlled studies exploring antiviral medication to treat ME/CFS [3,5,6].

### 4.4. Latent Virus–Host Immune Balance

Virgin et al. [40] has reviewed the intricate balance and synergy in the co-existence between latent viruses and the host, which, to a large extent, also involves epigenetic regulations both from the host and the virus side [24]. The antiviral responses could be dysfunctional by, for instance, blocking IFN by autoAbs to IFN-I, the release of neutralizing antiviral Abs, and the condition of the immune system such as other infections, severe stress, toxins, or trauma. An attenuation and decrease in the immune system is known to initiate the start of the latent-to-lytic viral program with the release of reactivated viral particles followed by worsening of the ME/CFS symptoms. Type I interferons are produced in response to viral infection as a key part of the innate immune response with potent antiviral, antiproliferative, and immunomodulatory properties. This cytokine binds a plasma membrane receptor made of IFNAR1 and IFNAR2 that is ubiquitously expressed and thus is able to act on virtually all body cells. A deficiency of IFN-I in the blood is thought to be a hallmark of severe COVID-19 and may provide a rationale for a combined therapeutic approach [41].

Dysfunctional IFN signaling underlies aberrant responses to infection and autoimmune diseases, including type I interferonpathies such as systemic lupus erythematosus (SLE) [42]. Multiple antiviral IFN-I dysregulations have been described, including the viral evasion of immune recognition by viral proteins antagonizing the stimulator of interferon genes (STING) pathway [43,44], dysregulated IFNA2 receptor by the formation of novel isoforms via transposon exonization [45], and autoAbs neutralizing IFN-I [21,27].

Recently, we reported that COVID-19 triggered not only high saliva anti-SARS-CoV-2 IgG, but also concomitant elevated saliva IgG against reactivated HAdV [10] and EBV [9] in ME/CFS patients compared to healthy controls. The main finding from the present pilot study is that ME/CFS patients release a significantly higher amount of reactivated viral copies of EBV in airway epithelium/sputum compared to HD, and that autoAbs to IFN-I are not elevated in mild/moderate conditions. Possibly there is an elevation in severe ME/CFS and the elderly, which has to be verified in larger validation cohorts.

### 4.5. Limitations of the Study

The number of participants in this study is small, and this research area demands further studies in larger cohorts. The presence of released herpesviruses, predominantly EBV in sputum, as observed here, is, however, supported by parallel studies showing reactivated EBV in blood [34], which underlines our recent studies that saliva anti-EBV and anti-HAdV IgG Abs are elevated in ME/CFS.

### 4.6. Future Perspectives and Conclusions

ME/CFS and long COVID share similarities in antiviral immune dysregulation, and intense biomarker and therapeutic explorative studies are much in focus [46]. It would be of interest to perform longitudinal studies on the shedding of herpesviruses and HAdV variation over time. This could possibly disclose a connection between increased virus shedding and the activation of previously clinically silent, persistent infection in lymphoid tissue [36,39,47], as well as any relation to symptom level. Clinical well-controlled studies including severe ME/CFS in antiviral therapy are much needed. For example, intravenous Brincidofovir (BCF i.v.) is effective against HAdV [48] and has activity against EBV, HCMV [49,50], and HHV6 [51]. In an ongoing Phase 2a clinical trial (NCT04706923), the antiviral activity of BCV i.v. is being evaluated in immunocompromised patients with HAdV viremia or disseminated HAdV disease. With this regime, viremia clearance was achieved in ten out of ten patients after a mean duration of treatment for 5.1 weeks [52].

To determine the role of HAdV and herpesviruses in the pathogenesis of ME/CFS, more extensive studies should be performed that include a larger number of participants. The most abundant sites of HAdV replication and infectious virus production are the intestine, the lower airway tract, and the eyes, where the virus can cause a variety of clinical manifestations ranging from mild to severe diseases [53]. Notably, prior to the onset of ME/CFS, a majority of patients report irritable bowel syndrome (IBS). In addition to intestinal lymphocytes, there are convincing findings showing that HAdV can establish persistent infections in tonsillar and adenoidal T-lymphocytes, in epithelium cells of the lung mucosa, and in brain tissue [36,47,54,55].

We have explored the release of reactivated latent viruses in airway mucosa by collecting sputum samples from 29 participants: ME/CFS patients, age-matched healthy controls, elderly healthy controls, and immunosuppressed active controls. We found that ME/CFS patients had a significantly higher viral copy number/mL of EBV compared to healthy donors. HHV6 was released in 50% of participants. HAdV was not found in patients, nor in controls, except for a young ME/CFS patient with severe symptoms and one of the participants treated with airway glucocorticoids. AutoAbs to type I IFN were not raised in the investigated ME/CFS cohort, apart from the very same patient that released HAdV and in the eldest 77-year-old participant.

## Figures and Tables

**Figure 1 viruses-17-00422-f001:**
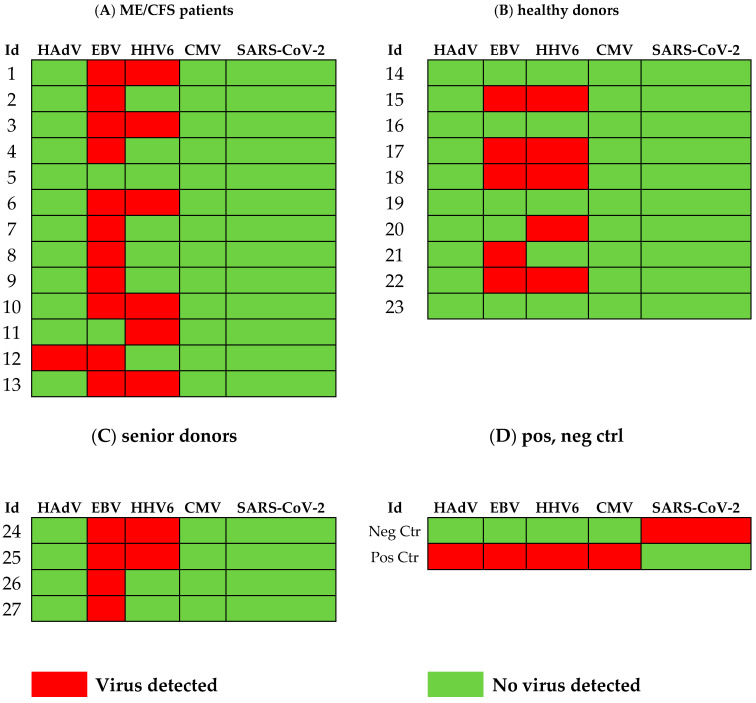
Overview of PCR-positive HAdV, EBV, HHV6, CMV, and SARS-CoV-2 in sputum from the 29 participants in the study. Red squares represent samples with virus present. Green squares represent samples with no virus detected.

**Figure 2 viruses-17-00422-f002:**
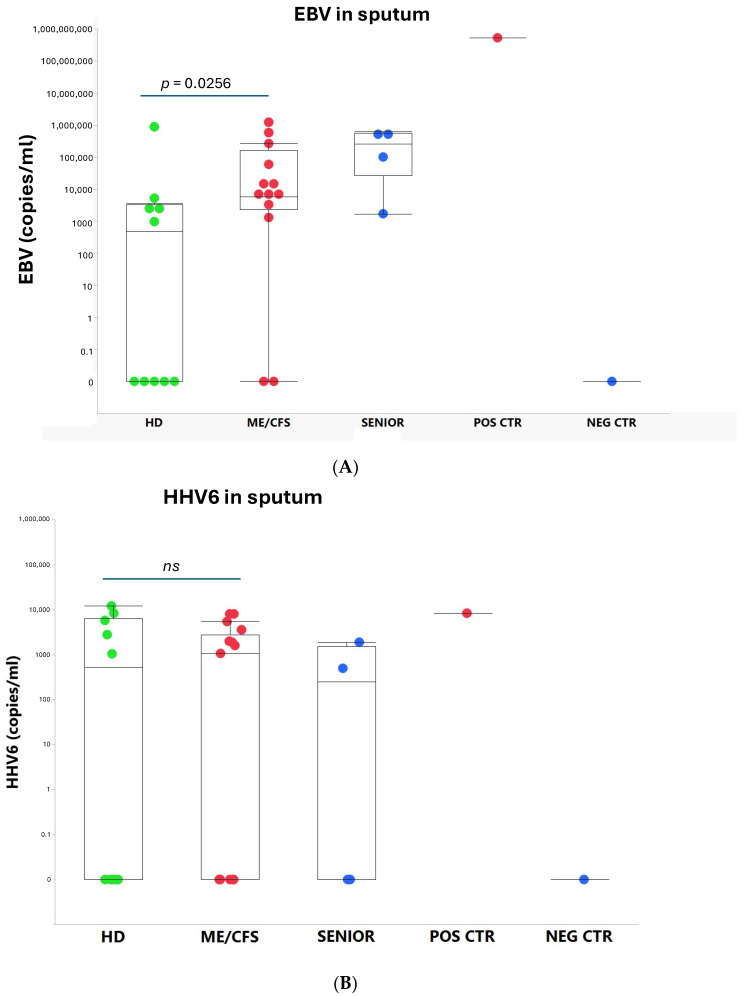
(**A**) EBV (copies/mL) in sputum from the five participant groups: HD = healthy donors (green dots, n = 10), ME/CFS patients (red dots, n = 13), SENIOR healthy controls (blue dots, n = 4), POS CTR = positive control, NEG CTR = negative control. Data are presented as boxplots with median values and outliers. Statistically significant difference was found between the HD and ME/CFS groups (*p* = 0.0256), and HD vs. SENIORS showed a trend (*p* = 0.0706) according to the non-parametric Wilcoxon rank procedure. (**B**) HHV6 (copies/mL) in sputum from the same participant groups, as shown in (**A**), was calculated. *p* < 0.05 indicates statistically significant differences. *ns*, non-significant difference.

**Figure 3 viruses-17-00422-f003:**
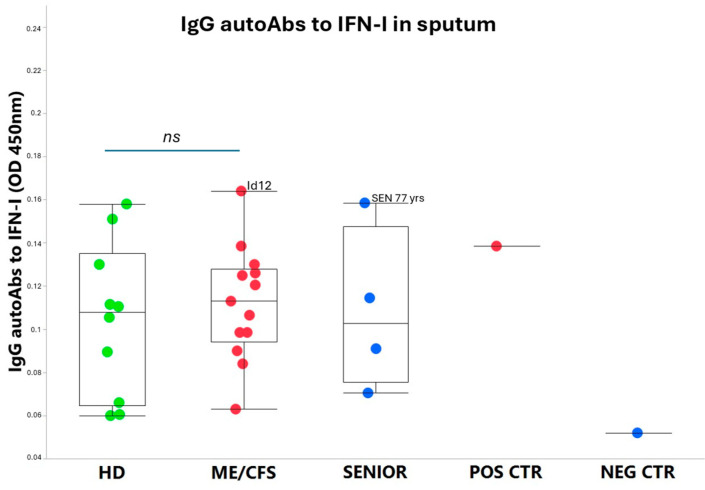
AutoAbs to type I interferon (IFN-I) in sputum (diluted 1:16) from the five participant groups. HD = healthy donors (green dots, n = 10), ME/CFS patients (red dots, n = 13), SENIOR healthy controls (blue dots, n = 4), POS CTR = positive control, NEG CTR = negative control. Data are presented as boxplots with median values and quartiles. Statistical significant difference in autoAb concentrations was not found (*ns*, non-significant) between the HD and ME/CFS groups according to a non-parametric Wilcoxon rank procedure. *p* < 0.05 indicates statistically significant differences. *ns*, non-significant difference.

**Table 1 viruses-17-00422-t001:** Sample Id, duration of ME/CFS, disease severity, sex, and age of the participants.

	ME/CFS Patients				Healthy Controls				Senior Controls			
Sample	Duration of	Disease	Sex	Age	Sample	Duration of	Sex	Age	Sample	Duration of	Sex	Age
Id	ME/CFS (yrs)	Severity		yrs	Id	ME/CFS (yrs)		yrs	Id	ME/CFS (yrs)		yrs
1	8	2	F	61	14	NA	F	56	24	NA	F	65
2	13	2	F	60	15	NA	F	33	25	NA	M	77
3	10	2	F	58	16	NA	F	61	26	NA	F	72
4	12	1	F	56	17	NA	F	61	27	NA	M	75
5	28	1	F	54	18	NA	F	37	Median			73.5
6	12	1	F	54	19	NA	F	49	Range			65–77
7	10	1	F	53	20	NA	F	66				
8	14	3	F	49	21	NA	M	48				
9	14	3	F	48	22	NA	F	33	**Positive, negative control**			
10	8	3	F	37	23	NA	F	66	**Sample**	**Duration of**	**Sex**	**Age**
11	17	2	F	37	Median			54	**Id**	**ME/CFS (yrs)**		**yrs**
12	12	3	F	22	Range			33–66	NEG CTR	NA	F	46
13	27	2	F	59					POS CTR	NA	F	54
Median	12			52.5								
Range	8- 28			22–61								

The participants were divided into four groups: ME/CFS patients (n = 13), healthy controls (n = 10), senior controls (n = 4), positive and negative controls (n = 2). Disease severity: 1 (mild): approximately 50% reduction in activity, 2 (moderate): mostly housebound,3 (severe): mostly bedbound, and 4 (very severe): bedbound and dependent on help for physical functions. NA: not applicable.

## Data Availability

All data generated or analyzed during this study are included in this published article.

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
