# Peer review of "Prevalence of EBV, HHV6, HCMV, HAdV, SARS-CoV-2, and Autoantibodies to Type I Interferon in Sputum from Myalgic Encephalomyelitis/Chronic Fatigue Syndrome Patients"

_viruses, 2025, doi:10.3390/v17030422_

Round 1
Reviewer 1 Report
Comments and Suggestions for Authors
This study is to investigate the relationship between the reactivation of latent viruses (EBV, HHV6 and HAdV) and the autoantibodies against IFNs in ME/CFS patients. The authors indicate that EBV level released by ME/CFS patients is higher than the healthy controls. And autoantibodies against IFN can’t explain the mechanism of the reactivation of the viruses. This manuscript is very clear, and this kind of study is very important for understanding the underlying mechanism of ME/CFS and long COVID. The main limitation of this study is that the sample size is too small and might not be representative for the whole population. Anyway, the authors mentioned this is a pilot study and I am looking forward to the future study with a large cohort. I have few minor comments for this manuscript, please see below.
- In abstract, a lot of “;” should be replaced with “,”.
- In line 15, the full name of EBV and HAdV should be clarified for the first time. The full terms are required to be clarified when the abbreviations show up for the first time to avoid confusion.
- In line 18, the full name of “autoAbs” should be clarified when it shows for the first time.
- In line 29, a full stop dot is missing.
- In line 36, please clarify the time when the data from CDC released. Besides, please add the reference of worldwide patient data source.
- In line 44, please include the references in one pair of square brackets.
- In line 57, please add the reference(s) for the sentence.
- In line 100, please replace “SARS-Cov-2” with “SARS-CoV-2” for consistency.
- The Figure 2A and 2B are separated, I would suggest the authors to integrate Figure 2A and 2B into a single figure.
- In line 195 and 268, please include the references in one pair of square brackets.
- In line 200, the full term “systemic lupus erythematosus (SLE)” should be clarified for the first time.
- In line 294, please replace “ME-CFS” with “ME/CFS” for consistency.
- In line 307, please clarify “BCV IV” as “intravenous Brincidofovir”.
- There are some extra spaces in the text should be deleted.
The English looks good, but with some flaws of writing mistakes.
Author Response
Response to reviewer #1.
Thank you for the appreciation of our brief report.
Our point-by-point response to specific comments. Line number refers to revised ms:
1. In abstract, a lot of “;” should be replaced with “,”. Our response: This is now done throughout the Abstract. Line 16-37.
2. In line 15, the full name of EBV and HAdV should be clarified for the first time. The full terms are required to be clarified when the abbreviations show up for the first time to avoid confusion. Our response: Done. Line 19-22.
3. In line 18, the full name of “autoAbs” should be clarified when it shows for the first time. Our response: Done. Line 24.
4. In line 29, a full stop dot is missing. Our response: Now added.
5. In line 36, please clarify the time when the data from CDC released. Besides, please add the reference of worldwide patient data source. Our response: The year was 2021-2022 and a proper reference was added and another for the worldwide patient data. Line 47-48.
6. In line 44, please include the references in one pair of square brackets. Our response: Done. Line 58.
7. In line 57, please add the reference(s) for the sentence. Our response: We have added 3 references for the sentence. Line 75.
8. In line 100, please replace “SARS-Cov-2” with “SARS-CoV-2” for consistency. Our response: Done. Line 154.
9. The Figure 2A and 2B are separated, I would suggest the authors to integrate Figure 2A and 2B into a single figure. Our response: Done. Line 214.
10. In line 195 and 268, please include the references in one pair of square brackets. Our response: Done. Note, one irrelevant ref (Ghia et al 2008) was removed. Line 293 and line 384.
11. In line 200, the full term “systemic lupus erythematosus (SLE)” should be clarified for the first time. Our response: Done. Line 298.
12. In line 294, please replace “ME-CFS” with “ME/CFS” for consistency. Our response: Done. Line 422.
13. In line 307, please clarify “BCV IV” as “intravenous Brincidofovir”. Our response: Done. Line 414.
14. There are some extra spaces in the text should be deleted. Our response: Done.
Reviewer 2 Report
Comments and Suggestions for Authors
1) Line 60-61: "several viruses...". Please be more explicit.
2) The introduction section is too short to generate a context that makes plausible the research.
3) The author reported the mean, but age in Table 1 did not have a normal distribution.
4) For DNA viruses, it is correct PCR, but for SARS-CoV-2, it was RT-qPCR.
5) In the EBV copies, was there not a difference between HD and senior?
6) Section 4.7: Convert it to a discussion section. There is more information to discuss.
Author Response
Response to Reviewer #2
Thank you for detailed and constructive suggestions, which we have followed. Line number refers to revised ms.
1) Line 60-61: "several viruses...". Please be more explicit. Our response: Now clarified. Line 89.
2) The introduction section is too short to generate a context that makes plausible the research. Our response: Now extended to generate a better context. Line 78-87.
3) The author reported the mean, but age in Table 1 did not have a normal distribution. Our response: Now corrected to median values.
4) For DNA viruses, it is correct PCR, but for SARS-CoV-2, it was RT-qPCR. Our response: Now clarified. Line 150-171.
5) In the EBV copies, was there not a difference between HD and senior? Our response: There was no significant difference, but a trend p = 0.0706. This information is now inserted into the Results and in the figure legend. Line 198 and line 222.
6) Section 4.7: Convert it to a discussion section. There is more information to discuss. Our response: Sections 4.1 to 4.7 are all Discussion sections, but we have moved section 4.1 (as advice by reviewer #3) and fused it with section 4.7 – now section 4.6 – the title and content are revised for improved clarity. Line 405-449.
Reviewer 3 Report
Comments and Suggestions for Authors
The manuscript reports the results of the study of reemerging of some herpesviruses (EBV, HHV, HCMV) and HAdV6 in myalgic encephalomyelitis/chronic fatigue syndrome (ME/CFS) patients. Based on sputum samples analysis (qPCR+ELISA), the authors revealed significant intergroup (healthy volunteers vs patients) differences only in EBV copy number/ml. At the same time, no statistically significant remanifestation of HHV6, CMV or HAdV6, as well as anti-IFNa antibodies were found in patients, except of one severe case and elderly control. Although this brief report is of certain virological value, a number of following flaws needs to be addressed.
First of all, the title should be rephrased, as HHV6 and HAdV appeared not to be reinactivated, and no increase in anti-IFNa antibodies was revealed. The 'viral evasion of IFN-I effect' mentioned in line 27 is unclear. By the way, the Abstract is strangely rich in semicolons.
Lines 34-36 and 57-60 should be referenced (not only by CDC mentioning). Moreover, the authors' aim to study the levels of anti-IFNa autoantibodies (stated in lines 62-63) is unclear from the Introduction. This topic should be explained in more detail.
'COVID' needs to be replaced by 'COVID-19' throughout the text.
The authors state that ME/CFS and long COVID-19 (or post-infection syndromes in general) are very difficult to distinguish for several times (e.g., in lines 49, 196-198, 285-286). Given this fact, it's necessary to clearly explain the ME/CFS diagnosing in participants (lines 66-67) and key hallmarks of ME/CFS.
The inclusion criteria for healthy control group should also be given.
The designations NEG CTR and POS CTR, which were introduced in Materials and Methods, are absent in Table 1 and Figure 1 (which in turn contain P7 and L59, that's confusing).
The severity criteria (Table 1) should be described.
How was a saliva contamination avoided (line 92)?
The qPCR parameters (standards & calibration, threshold cycle, software & detection system, etc.) should be given.
In line 101, DNA extraction is mentioned, while SARS-CoV-2 is an RNA-containing virus.
Line 114: may be TMB is implied?
The Materials and Methods section lacks the description of statistical data processing (p-level, software, tests, normality check, etc.)
'The highest EBV load' should be specified in line 122.
Why nothing is mentioned about SARS-CoV-2 positivity of P7 participant (Figure 1)?
In Figure 2A, the differences between POS CTR and HD are considered to be insignificant.
In lines 139-141 Figure 1 should be mentioned.
Lines 163-164: given the overall modest OD450 values (Figure 3), 'the highest level' seems to be a speculation.
Line 166: Why all the participants turned to be positive for anti-type-I IFN IgG autoAbs? The normality of this phenomenon in healthy donors should be explained.
In addition to OD450 values, AUCs should be given in Figure 3 to access the titers of autoAbs.
Lines 172-174 are similar to the lines 149-151.
Lines 176-185 look like a conclusions, so need to be moved as a separate paragraph to the end of the paper accompanied with a shortened future prospectives. In the current version, the Discussion seems to be incomplete.
Line 187: what 'action' is implied?
Line 205: why anti-EBV Abs absence serves as a sign of increased viral production?
Lines 211-212: a very small sample size was studied here to judge about the HAdV releasing in severe patients.
Lines 217-222: more papers studying EBV reactivation in ME/CFS patients need to be reviewed.
Line 227: why frequency of HAdV infection in patients with 'airway symptoms' is considered instead of ME/CFS affected? The articles implying HAdV activation in immunocompromised people should be in line with the manuscript topic.
Lines 245-247 should be referenced.
Line 270: what is implied as 'COVID-19 triggered higher saliva anti-SARS-CoV-2 IgG'? This statement seems to be senseless.
Line 273: 'viral copies' should be specified (which virus is implied).
Lines 303-309: it would be better to give the description of brincidofovir when it was first mentioned (in lines 291-292).
Finally, the paper contains some grammar/stylistic flaws (e.g., 'infections trauma' in line 52, 'is' in line 78, 'not known' in line 201, 'study' duplication in line 225, 'antivirus IgG' in line 281), so a careful proofreading is adviced for the authors.
Author Response
Response to reviewer #3
Many thanks for carefully reading our paper and giving constructive suggestions for improvement and clarity. Line number refers to revised ms.
First of all, the title should be rephrased, as HHV6 and HAdV appeared not to be reinactivated, and no increase in anti-IFNa antibodies was revealed. The 'viral evasion of IFN-I effect' mentioned in line 27 is unclear. By the way, the Abstract is strangely rich in semicolons. Our response: The title is now modified. ‘…viral evasion…’ in Abstract is clarified and semicolons removed.
Lines 34-36 and 57-60 should be referenced (not only by CDC mentioning). Moreover, the authors' aim to study the levels of anti-IFNa autoantibodies (stated in lines 62-63) is unclear from the Introduction. This topic should be explained in more detail. Our response: CDC ref now inserted. The aim of the study is now better explained in the Introduction. Line 47 and lines 78-87.
'COVID' needs to be replaced by 'COVID-19' throughout the text. Our response: post-SARS-CoV-2 syndrome is termed long COVID by authorities such as NIH and WHO, albeit long COVID-19 can also be used. We prefer the shorter version.
The authors state that ME/CFS and long COVID-19 (or post-infection syndromes in general) are very difficult to distinguish for several times (e.g., in lines 49, 196-198, 285-286). Given this fact, it's necessary to clearly explain the ME/CFS diagnosing in participants (lines 66-67) and key hallmarks of ME/CFS. Our response: We have now explained the diagnosing in section 2.1. Line 98-114.
The inclusion criteria for healthy control group should also be given. Our response: Inclusion criteria for HD group now inserted in section 2.1. Line 105-107.
The designations NEG CTR and POS CTR, which were introduced in Materials and Methods, are absent in Table 1 and Figure 1 (which in turn contain P7 and L59, that's confusing). Our response: We have now inserted POS CTR and NEG CTR in Table 1 and Figure 1 for clarity.
The severity criteria (Table 1) should be described. Our response: This is now added in Materials and Methods, section Participants, and in the Table 1 legend. Line 110-114 and line 132.
How was a saliva contamination avoided (line 92)? Our response: We have modified the sentence and included: The detailed instructions for sputum collection included twice rinsing mouth with water. The mucus sputum sample was then collected by strong coughing of sputum into a sterile 50 ml plastic tube, trying to avoid saliva contamination. This is now added to section 2.3. Line 143-145.
The qPCR parameters (standards & calibration, threshold cycle, software & detection system, etc.) should be given. Our response: The details are now inserted into section 2.4. Line 159-171.
In line 101, DNA extraction is mentioned, while SARS-CoV-2 is an RNA-containing virus. Our response: Nucleic acids (DNA and RNA) were prepared using the same kit, which is now explained in detail. Line 156.
Line 114: may be TMB is implied? Our response: Yes, thank you, this misprint is now corrected. Line 184.
The Materials and Methods section lacks the description of statistical data processing (p-level, software, tests, normality check, etc.). Our response: Now added as section 2.6. Line 187-191.
'The highest EBV load' should be specified in line 122. Our response: The highest EBV load was 526 million copies/ml. This is now inserted in section 3.1. Line 199.
Why nothing is mentioned about SARS-CoV-2 positivity of P7 participant (Figure 1)? Our response: Now inserted in section 3.1. Line 205-207.
In Figure 2A, the differences between POS CTR and HD are considered to be insignificant. Our response: No statistics were calculated since POS CTR is one donor only.
In lines 139-141 Figure 1 should be mentioned. Our response: Now inserted. Line 235.
Lines 163-164: given the overall modest OD450 values (Figure 3), 'the highest level' seems to be a speculation. Our response: Yes, we have revised accordingly. Healthy donors have a low basic level of autoAbs to IFN-I, and the Abs rise during acute infections particularly in severely ill COVID-19 patients and in autoimmune diseases (ref Hale 2023 in the MS). Sputum samples were diluted 1:16, as explained in section 2.5, which gives modest OD450 values. This information is inserted in Figure 3 legend. Importantly, we agree with the reviewer that ‘the highest level’ should be modified. We have revised accordingly in section 3.2. Line 152-158. Line 161.
Line 166: Why all the participants turned to be positive for anti-type-I IFN IgG autoAbs? The normality of this phenomenon in healthy donors should be explained. Our response: All participants expressed a low but detectable basic level of autoAbs to IFN-I, in comparison with the negative control donor that had very low level of Abs overall, due to anti-CD19 (Rituximab) therapy, thus expressing background level. The healthy donor anti-IFN-I basic level has been found in multiple studies (for review see Hale 2023), however 4% of healthy elderly donors do express significantly increased levels (ref Bastard et al. Sci Immunol 2021) see Discussion section 4.2. Line 296-298.
In addition to OD450 values, AUCs should be given in Figure 3 to access the titers of autoAbs. Our response: The suggested expansion including titration of autoAbs and AUC determination would be very interesting and would be meaningful when compared with cohorts of severe/very severe ME/CFS patients or acutely infected patients with COVID-19 or EBV-triggered infectious mononucleosis. However, considering that both HD and ME/CFS groups in our present pilot study expressed low, basic levels of autoAbs to IFN-I, such an extension would require a study in its own.
Lines 172-174 are similar to the lines 149-151. Our response: This is now corrected. Line 265.
Lines 176-185 look like a conclusions, so need to be moved as a separate paragraph to the end of the paper accompanied with a shortened future prospectives. In the current version, the Discussion seems to be incomplete. Our response: We have now revised accordingly and moved the section to the end, and shortened future perspectives. Line 440-449.
Line 187: what 'action' is implied? Our response: We have simplified the title of Discussion section 4.1 (previous section 4.2) and removed the word ‘action’. Line 284.
Line 205: why anti-EBV Abs absence serves as a sign of increased viral production? Our response: This sentence refers to the study by Apostolou et al 2022, showing that ME/CFS participants have higher anti-EBV levels than HD. This finding was interpreted as a more frequent release of reactivated EBV in ME/CFS resulting in increased anti-EBV production. We have clarified these sentences. Line 304-305.
Lines 211-212: a very small sample size was studied here to judge about the HAdV releasing in severe patients. Our response: We agree that we cannot draw any conclusion based on the single severe ME/CFS case. We have revised the text accordingly. Line 312-315.
Lines 217-222: more papers studying EBV reactivation in ME/CFS patients need to be reviewed. Our response: We have included additional references for EBV reactivation in ME/CFS. Line 321-322.
Line 227: why frequency of HAdV infection in patients with 'airway symptoms' is considered instead of ME/CFS affected? The articles implying HAdV activation in immunocompromised people should be in line with the manuscript topic. Our response: We agree that HAdV activation in immunocompromised persons is in line with manuscript topic. Our results show that HAdV activation is infrequent and most likely associated with immunosuppression: The POS CTR donor was immunocompromised by glycocorticoids and HAdV positive. The Id12 patient with severe ME/CFS had active HAdV possibly due to immunosuppression/exhaustion resulting in dysfunctional antiviral surveillance. The Discussion section 4.3 is revised according to reviewer’s suggestion. Line 331-336.
Lines 245-247 should be referenced. Our response: References now added. Line 361.
Line 270: what is implied as 'COVID-19 triggered higher saliva anti-SARS-CoV-2 IgG'? This statement seems to be senseless. Our response: Yes, it goes without saying that COVID-19 gives anti-SARS-CoV-2 IgG – at least in healthy persons. We have revised the sentence. Line 387-388.
Line 273: 'viral copies' should be specified (which virus is implied). Our response: Now specified. Line 391.
Lines 303-309: it would be better to give the description of brincidofovir when it was first mentioned (in lines 291-292). Our response: Done. Line 414.
Finally, the paper contains some grammar/stylistic flaws (e.g., 'infections trauma' in line 52, 'is' in line 78, 'not known' in line 201, 'study' duplication in line 225, 'antivirus IgG' in line 281), so a careful proofreading is adviced for the authors. Our response: Done.
Reviewer 4 Report
Comments and Suggestions for Authors
The authors suggested using a well-known research method. This recommendation can be accepted because it does not contradict existing practice. I did not find any critical shortcomings or contradictions. I believe that the article can be published in its original version.
Author Response
Reviewer #4 response. We are very grateful for reviewer’s appreciation of our paper.
Round 2
Reviewer 2 Report
Comments and Suggestions for Authors
The manuscript has been improved, enougth to be accepted.